# TENSOR RING NETS
# ADAPTED DEEP MULTI-TASK LEARNING

## ABSTRACT

Recent deep multi-task learning (MTL) has been witnessed its success in alleviating data scarcity of some task by utilizing domain-specific knowledge from related tasks. Nonetheless, several major issues of deep MTL, including the effectiveness of sharing mechanisms, the efficiency of model complexity and the flexibility of network architectures, still remain largely unaddressed. To this end, we propose a novel generalized latent-subspace based knowledge sharing mechanism for linking task-specific models, namely tensor ring multi-task learning (TRMTL). TRMTL has a highly compact representation, and it is very effective in transferring task-invariant knowledge while being super flexible in learning task-specific features, successfully mitigating the dilemma of both negative-transfer in lower layers and under-transfer in higher layers. Under our TRMTL, it is feasible for each task to have heterogenous input data dimensionality or distinct feature sizes at different hidden layers. Experiments on a variety of datasets demonstrate our model is capable of significantly improving each single task's performance, particularly favorable in scenarios where some of the tasks have insufficient data.

## 1 INTRODUCTION

Multi-task learning (MTL) (Caruana, 1997; Maurer et al., 2016) is an approach for boosting the overall performance in each individual task by learning multiple related tasks simultaneously. In the deep learning context, jointly fitting sufficiently flexible deep neural networks (DNNs) to data of multiple tasks can be seen as adding an inductive bias to the deep models, which could be beneficial to learn feature representations preferable by all tasks. Recently, the deep MTL has gained much popularity and been successfully explored in an abroad range of applications, such as computer vision (Zhang et al., 2014; Misra et al., 2016), natural language processing (Luong et al., 2015; Liu et al., 2017), speech recognition (Wu et al., 2015; Huang et al., 2015) and so on.

However, a number of key challenges posed by the issues of *ineffectiveness*, *inefficiency* and *inflexibility* in deep MTL are left largely unaddressed. One major challenge is how to seek effective information sharing mechanisms across related tasks, which is equivalent to designing better parameter sharing patterns in the deep networks. Some previous work (Zhang et al., 2014; Yin & Liu, 2017) tried to solve this problem by means of hard parameter sharing (Ruder, 2017), where the bottom layers are all shared except with one branch per task at the top layers. Although being simple and robust to over-fitting (Baxter, 1997), this kind of architecture can be harmful when learning high-level task-specific features, since it focuses only on common low-level features of all tasks. Moreover, these common features may be polluted by some noxious tasks, leading to the negative transfer in low-level features among tasks (Yosinski et al., 2014). An alternative line of work mitigate this issue to some extent by following the soft parameter sharing strategy (Ruder, 2017), under which one separate DNN is learned for each task with its own set of parameters, and the individual DNNs are implicitly linked by imposing constraints on the aligned weights. The deep MTL models of this type include using $\ell_2$ norm regularization (Duong et al., 2015), trace norm regularization (Yang & Hospedales, 2016) and tensor norm priors (Long & Wang, 2015; Long et al., 2017).

The lack of efficiency in model complexity gives rise to another great challenge for current deep MTL. The above soft-sharing based deep models (one set of parameters per task) typically involve enormous number of trainable parameters and require extremely large storage and memory. It is thus usually infeasible to deploy those deep MTL models on resource-constrained devices such as mobile

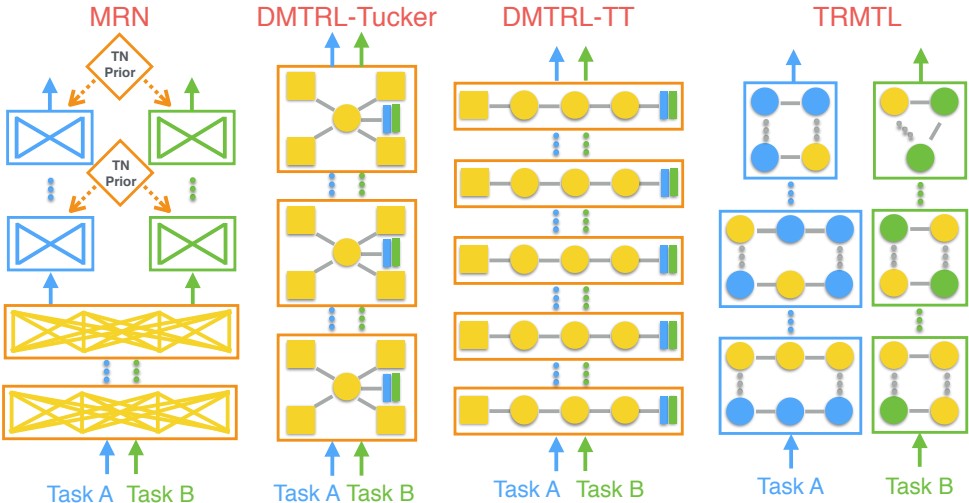

Figure 1: The overall sharing mechanisms of MRN, two variants of DMTRL (for the setting of CNN) and our TRMTL w.r.t. two tasks. The shared portion is depicted in yellow. The circles, squares and thin rectangles represent tensor cores, matrices and vectors, respectively. MRN: original weights are totally shared at the lower layers and the relatedness between tasks at the top layers is modeled by tensor normal priors. DMTRL (TT or Tucker): all layer-wise weights must be equal-sized so as to be stacked and decomposed into factors. For each task, almost all the factors are shard at each layer except the very last 1D vector. Such pattern of sharing is identical at all layers. TRMTL: layer-wise weights are separately encoded into TR-formats for different tasks, and a subset of latent cores are selected to be tied across two tasks. The portions of sharing can be different from layer to layer.

phones and wearable computers. Yang & Hospedales (2017) alleviated the issue by integrating tensor factorization with deep MTL and proposed deep multi-task representation learning (DMTRL). Specifically, they first stack up the layer-wise weights from all tasks and then decompose them into low-rank factors, yielding a succinct deep MTL model with fewer parameters. Despite the compactness of the model, DMTRL turns out to be rather restricted on sharing knowledge effectively. This is because, as shown in Figure 1, DMTRL (TT or Tucker) shares almost all fractions of layer-wise weights as common factors, leaving only a tiny portion of weights to encode the task-specific information. Even worse, such pattern of sharing must be identical across all hidden layers, which is vulnerable to the negative transfer of the features. As an effect, the common factors become highly dominant at each layer and greatly suppress model's capability in expressing task-specific variations.

The last challenge arises from the flexibility of architecture in deep MTL. Most of deep MTL models force tasks to have the equal-sized layer-wise weights or input dimensionality. This restriction makes little sense for the case of loosely-related tasks, since individual tasks' features (input data) can be quite different and the sizes of layer-wise features (input data) may vary a lot from task to task.

In this work, we provide a generalized latent-subspace based solution to addressing aforementioned difficulties of deep MTL, from all aspects of *effectiveness*, *efficiency* and *flexibility*. Regarding the effectiveness, we propose to share different portions of weights as common knowledge at distinct layers, so that each individual task can better convey its private knowledge. As for the efficiency, our proposal shares knowledge in the latent subspace instead of original space by utilizing a general tensor ring (TR) representation with a sequence of latent cores (Zhao et al., 2016; 2017). One motivation of TR for MTL is it generalizes other chain structured tensor networks (Cichocki et al., 2016), especially tensor train (TT) (Oseledets, 2011), in terms of model expressivity power, as TR can be formulated as a sum of TT networks. On the other hand, TR is able to approximate tensors using lower overall ranks than TT does (Zhao et al., 2016), thus yielding a more compact and sparsely-connected model with significantly less parameters for deep MTL. Adopting TR-format with much lower ranks could bring more benefits to deep MTL if we tensorize a layer-wise weight of each task into a higher-order weight tensor, since the weight can be decomposed into a relatively larger number but smaller-sized cores. This in turn facilitates the sharing of cores at a finer granularity and further enhances the effectiveness of sharing. Additionally, Zhao et al. (2017) observed that different cores control different levels of correlations in tensor data, e.g. for a tensorized image, each core

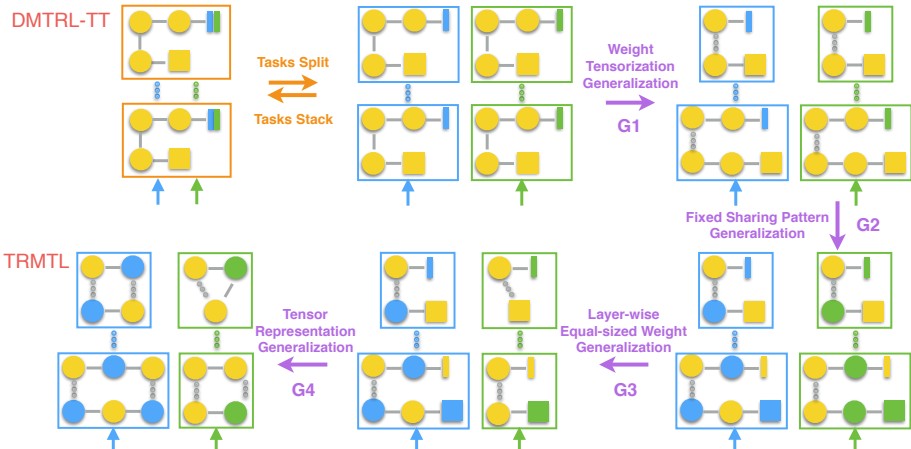

Figure 2: The demonstration of four types generalizations from DMTRL-TT to our TRMTL.

affects one specific scale of the resolution. Such observation also provides a natural inspiration for encoding weights of deep MTL via chain based tensor networks and then sharing the cores across tasks. For the last challenge, the flexibility of deep MTL networks is maximally retained in our proposal by parameterizing one DNN per task, while the discrepancy between tasks' features is also taken into account by encoding layer-wise weights of different tasks using distinct number of cores. In this way, the network of each task may possess its own size of weight (or input dimensionality). We refer to our framework as tensor ring multi-task learning (TRMTL), as depicted in Figure 1. With above properties, TRMTL achieves the state-of-the-art performance on a variety of datasets and we validate that each individual task can gain much benefit from the proposed architecture.

## 2 RELATED WORK

The classical matrix factorization based MTL (Kumar & Daume III, 2012; Romera-Paredes et al., 2013; Wimalawarne et al., 2014) requires the dimensionality of weight vectors $\{\mathbf{w}_t \in \mathbb{R}^M\}_{t=1}^T$ of $T$ tasks to be equal-sized, so that these weights could be stacked up into one weight matrix $\mathbf{W} \in \mathbb{R}^{M \times T}$. Kumar & Daume III (2012) assumes $\mathbf{W}$ to be low-rank and factorizes it as $\mathbf{W} = \mathbf{LS}$. Here, $\mathbf{L} \in \mathbb{R}^{M \times K}$ consists of $K$ task-independent latent basis vectors, whereas each column vector of $\mathbf{S} \in \mathbb{R}^{K \times T}$ is task-specific and contains the mixing coefficients of these common latent bases. (Yang & Hospedales, 2017) extended this matrix based MTL to its tensorial counterpart DMTRL by making use of tensor factorization. Likewise, DMTRL starts by putting the equal-sized weight matrices $\{\mathbf{W}_t \in \mathbb{R}^{M \times N}\}_{t=1}^T$ side by side along the 'task' mode to form a 3rd-order weight tensor $\mathcal{W} \in \mathbb{R}^{M \times N \times T}$. In the case of CNN, this weight tensor corresponds to a 5th-order filter tensor $\mathcal{K} \in \mathbb{R}^{H \times W \times U \times V \times T}$. DMTRL then factorizes $\mathcal{W}$ (or $\mathcal{K}$), for instance via TT-format, into 3 TT-cores (or 5 TT-cores for $\mathcal{K}$) (Yang & Hospedales, 2017). Analogously, the first 2 TT-cores (or the first 4 TT-cores) play exactly the same role as $\mathbf{L}$ for the common knowledge; the very last TT-core is in fact a matrix (similar to $\mathbf{S}$), with each column representing the task-specific information.

Our TRMTL differs widely with DMTRL and generalizes DMTRL from a variety of aspects. In order to reach TRMTL from DMTRL-TT, one needs to take *four major types of generalizations* (G1-G4), as demonstrated in Figure 2. Firstly (in G1), TRMTL tensorizes the weight into a higher-order weight tensor before factorizing it. By doing so, the weight can be embedded into more latent cores than that of just 3 cores (or 5 cores) in DMTRL, which yields a more compact model and makes the sharing at a finer granularity feasible. Secondly (in G2), DMTRL stringently requires that the first $D$-1 cores ($D$ is weight tensor's order) must be all shared at every hidden layer, only the last vector is kept for private knowledge. By contrast, TRMTL allows for any sharing pattern at distinct layer. Thirdly (in G3), there is no need for layer-wise weights to be equal-sized and stacked into one big tensor as in TRMTL, each task may have its individual input dimensionality. Finally (in G4), TRMTL further generalizes TT to TR-format. For each task in DMTRL, the first core must be a matrix and the last core must be a vector (with both border rank and outer mode size being 1). Notice that our TRMTL also conceptually subsumes DMTRL-Tucker in terms of the first three aspects of generalizations (G1-G3). It is also worth mentioning that Wang et al. (2018) only applies

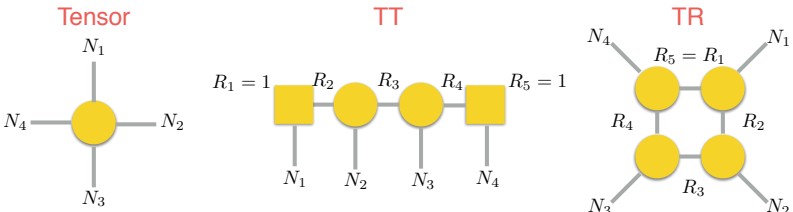

Figure 3: The diagrams of a 4th-order tensor and its TT-format and TR-format.

TR-format for weight compression in a single deep net, whereas ours incorporates a more general tensor network into the deep MTL context. The two methods differ in goals and applications.

Long et al. (2017) lately proposed MRN which incorporates tensor normal priors over the parameter tensors of the task-specific layers. MRN jointly learns the transferable features as well as multilinear relationship among tasks, with the objective to alleviate both under-transfer and negative-transfer of the knowledge. However, like methods (Zhang et al., 2014; Ouyang et al., 2014; Chu et al., 2015), MRN follows the architecture where all the lower layers are shared, which may harm the transferability if tasks are loosely correlated. In addition, the relatedness of tasks is captured by the covariance structures over features, classes and tasks. Constantly updating these covariance matrices (via SVD in (Long et al., 2017)) becomes computationally prohibitive for large scale networks. Compared to above mentioned *non-latent-subspace* methods, TRMTL is highly compact and hence needs much fewer parameters, which is obviously advantageous in tasks with small sample size.

## 3 TENSOR PRELIMINARIES

High-order tensors (Kolda & Bader, 2009) are referred to as multi-way arrays of real numbers. Let $\mathcal{W} \in \mathbb{R}^{N_1 \times \cdots \times N_D}$ be a $D$th-order tensor in calligraphy letter, where $D$ is called mode or way. Some very original work have successfully applied tensor decompositions to applications such as imaging analysis (Vasilescu & Terzopoulos, 2002; 2003) and computer vision (Vasilescu, 2011). A recent tensor ring decomposition (TR) (Zhao et al., 2016) decomposes a tensor $\mathcal{W}$ into a sequence 3rd-order latent cores that are multiplied circularly. An example of TR-format is illustrated in Figure 3. In TR-format, any two adjacent latent cores are 'linked' by a common dimension of size $R_{k+1}$, $k \in \{1, ..., D\}$. In particular, the last core is connected back to the first core by satisfying the border rank condition $R_{D+1} = R_1$. The collection of $[R_1, R_2, ..., R_D]$ is defined as TR-rank. Under TR-format, merely $\sum_{k=1}^{D} N_k R_k R_{k+1}$ parameters are needed to represent the original tensor $\mathcal{W}$ of size $\prod_{k=1}^{D} N_k$. Compared with TT-format (Oseledets, 2011), TR generalizes TT by relaxing the border rank condition. Zhao et al. (2016) conclude that TR is more flexible than TT w.r.t. low-rank approximation. The authors observe the pattern of ranks distribution on cores tend to be fixed in TT. In TT, the ranks of middle cores are often much larger than those of the side cores, while TR-ranks has no such drawbacks and can be equally distributed on cores. The authors also claim that, under the same approximation accuracy, the *overall ranks* in TR are usually much smaller than those in TT, which makes TR a more compact model than TT. For more favorable properties, such as TR is invariant under circular dimensional permutation, we refer readers to (Zhao et al., 2016; 2017).

## 4 METHODOLOGY

In general, our tensor ring multi-task learning (TRMTL) learns one DNN per task by representing the original weight of each layer with a tensor ring layer (TRL), i.e., utilizing a sequence of TR-cores. Then, a subset of TR-cores are tied across multiple tasks to encode the task-independent knowledge, while the rest TR-cores of each task are treated as private cores for task-specific knowledge.

### 4.1 TENSOR RING LAYER

We start the section by describing the tensor ring layer (TRL), which lays a groundwork for our TR based deep MTL approach. Following the TT-matrix (Novikov et al., 2015) representation, TR is able to represent a large matrix more compactly via TR-matrix format. Specifically, let **W**

be a matrix of size $M \times N$ with $M = \prod_{k=1}^{D} M_k, N = \prod_{k=1}^{D} N_k$, which can be reshaped into a $D$th-order tensor $\mathcal{W} \in \mathbb{R}^{M_1 N_1 \times M_2 N_2 \cdots \times M_D N_D}$ via bijective mappings $\phi(\cdot)$ and $\psi(\cdot)$. The map $\phi(i) = (\phi_1(i), ..., \phi_D(i))$ transforms the row index $i \in \{1, ..., M\}$ into a $D$-dimensional vector index $(\phi_1(i), ..., \phi_D(i))$ with $\phi_k(i) \in \{1, ..., M_k\}$; similarly, the map $\psi(\cdot)$ converts the column index $j \in \{1, ..., N\}$ also into a $D$-dimensional vector index $(\psi_1(j), ..., \psi_D(j))$ where $\psi_k(j) \in \{1, ..., N_k\}$. In this way, one can establish a one-to-one correspondence between a matrix element $\mathbf{W}(i, j)$ and a tensor element $\mathcal{W}((\phi_1(i), \psi_1(j)), ..., (\phi_D(i), \psi_D(j)))$ using the compound index $(\phi_k(\cdot), \psi_k(\cdot))$ for mode $k \in \{1, ..., D\}$. We formulate the TR-matrix format as

$$
\begin{aligned}
\mathbf{W}(i, j) &= \mathcal{W}((\phi_1(i), \psi_1(j)), ..., (\phi_D(i), \psi_D(j))) \\
&= \sum_{r_1=1}^{R_1} \sum_{r_2=1}^{R_2} \cdots \sum_{r_D=1}^{R_D} \mathcal{G}^{(1)}_{r_1,(\phi_1(i),\psi_1(j)),r_2} \mathcal{G}^{(2)}_{r_2,(\phi_2(i),\psi_2(j)),r_3} \cdots \mathcal{G}^{(D)}_{r_D,(\phi_D(i),\psi_D(j)),r_1} \\
&= \sum_{r_1=1}^{R_1} \mathbf{g}^{(1)\mathsf{T}}_{r_1}[(\phi_1(i), \psi_1(j))] \mathbf{G}^{(2)}[(\phi_2(i), \psi_2(j))] \cdots \mathbf{g}^{(D)}_{r_1}[(\phi_D(i), \psi_D(j))] \\
&= \text{Trace}\{\mathbf{G}^{(1)}[(\phi_1(i), \psi_1(j))] \mathbf{G}^{(2)}[(\phi_2(i), \psi_2(j))] \cdots \mathbf{G}^{(D)}[(\phi_D(i), \psi_D(j))]\}.
\end{aligned}
\tag{1}
$$

where $\mathcal{G}^{(k)}$ denotes the $k$th latent core, while $\mathbf{G}^{(k)}[(\phi_k(i), \psi_k(j))] \in \mathbb{R}^{R_k \times R_{k+1}}$ corresponds to the $(\phi_k(i), \psi_k(j))$th slice matrix of $\mathcal{G}^{(k)}$. $\mathbf{g}^{(1)\mathsf{T}}_{r_1}[(\phi_1(i), \psi_1(j))]$ represents the $r_1$th row vector of the $\mathbf{G}^{(1)}[(\phi_1(i), \psi_1(j))]$ and $\mathbf{g}^{(D)}_{r_1}[(\phi_D(i), \psi_D(j))]$ is the $r_1$th column vector of $\mathbf{G}^{(D)}[(\phi_D(i), \psi_D(j))]$. Notice that the third line in equation 1 implies TRL is more powerful than TT based layer in terms of the modeling expressivity, as TRL can in fact be written as a sum of $R_1$ TT layers. In the deep MTL context, the benefits of tensorization in our TRL are twofold: a sparser, more compact tensor network format for each task and a potentially finer sharing granularity across the tasks.

With TRL, the training can be conducted by applying the standard stochastic gradient descent based methods on the cores. Note that TRL is similar to the recently proposed TR based weight compression (Wang et al., 2018) for neural network, but we adopt a different 4th-order latent cores in TR-matrix. As for CNN setting, one can easily extend TR to a convolutional kernel $\mathcal{K} \in \mathbb{R}^{H \times W \times U \times V}$

$$
\mathcal{K}(h, w, u, v) = \text{Trace}\{\mathbf{G}^{(0)}[(h, w)] \mathbf{G}^{(1)}[(\phi_1(u), \psi_1(v))] \cdots \mathbf{G}^{(D)}[(\phi_D(u), \psi_D(v))]\}. \tag{2}
$$

## 4.2 TENSOR RING MULTI-TASK LEARNING

Our sharing strategy is to partition each layer's parameters into task-independent TR-cores as well as task-specific TR-cores. More specifically, for some hidden layer of an individual task $t \in \{1, ..., T\}$, we begin with reformulating the layer's weights $\mathbf{W}_t \in \mathbb{R}^{U_t \times V_t}$ in terms of TR-cores by means of TRL, where $U_t = \prod_{k=1}^{D_t} U_t^k, V_t = \prod_{k=1}^{D_t} V_t^k$. We thereafter reshape a layer's input $\mathbf{h}_t \in \mathbb{R}^{U_t}$ into a $D_t$th-order tensor $\mathcal{H}_t \in \mathbb{R}^{U_t^1 \times \cdots \times U_t^{D_t}}$. Next, the layer's input tensor $\mathcal{H}_t$ can be transformed into layer's output tensor $\mathcal{Y}_t \in \mathbb{R}^{V_t^1 \times \cdots \times V_t^{D_t}}$ via $\mathcal{W}_t$ in TR-format. Finally, we have

$$
\begin{aligned}
\mathcal{Y}_t(v_1, ..., v_{D_t}) = \sum_{u_1=1}^{U_1} \cdots \sum_{u_{D_t}=1}^{U_{D_t}} \mathcal{H}_t(u_1, ..., u_{D_t}) \text{Trace}\{\mathbf{G}^{(1)}_{com}[(u_1, v_1)] \\
\cdots \mathbf{G}^{(p)}_t[(u_p, v_p)] \cdots \mathbf{G}^{(q)}_{com}[(u_q, v_q)] \cdots \mathbf{G}^{(r)}_t[(u_r, v_r)] \cdots \mathbf{G}^{(D_t)}_{com}[(u_{D_t}, v_{D_t})]\},
\end{aligned}
\tag{3}
$$

where the common TR-cores subset $\{\mathbf{G}^{(\cdot)}_{com}\}$ has $c$ elements which can be arbitrarily chosen from the set of all $D_t$ cores, leaving the rest cores $\{\mathbf{G}^{(\cdot)}_t\}$ as task-specific TR-cores. Pay close attention that our TRMTL neither restricts on which cores to share, nor restricts the shared cores to be in an consecutive order. Finally, we reshape tensor $\mathcal{Y}_t$ back into a vector output $\mathbf{y}_t \in \mathbb{R}^{V_t}$. Note that the portion of sharing, which is mainly measured by c, can be set to different values from layer to layer. According to equation 3, TRMTL represents each weight element in weight matrix as function of a sequence product of the slice matrices of the corresponding shared cores and private cores. Intuitively, this strategy suggests the value of each weight element is partially determined by some common latent factors, and meanwhile, also partially affected by some private latent factors. Thus, our sharing is carried out in an distributed fashion. This is more efficient than conventional sharing strategies in which each weight element is either $100\%$ shared or $100\%$ not shared.

### 4.3 REMARKS ON CORE SELECTION

There are various strategies on how to select the shard cores w.r.t. both the location and the number. Zhao et al. (2017) find that distinct cores control an image at different scales of resolution. The authors demonstrate this by decomposing a tensorized 2D image into TR-cores, and then adding noise to one specific core at a time. They show the core in the first location controls small-scale patches while the core in the last location influences on large-scale partitions. Motivated by this, in current work, we preferentially share the features from the detailed scale to the coarse scale, which means we follow a natural left-to-right order in location to select different $c$ number of cores at distinct layers. A more sophisticated and possible option is to automatically select sharable core pairs that have highest similarity. We may consider two cores as a candidate pair if the same perturbation of the two cores induces similar changes in the errors of respective tasks. In this way, one can adaptively select most similar cores from tasks according to a certain threshold, leaving the rest as private cores.

## 5 EXPERIMENTAL RESULTS

We compare our TRMTL with single task learning (STL), MRN (Long et al., 2017), two variants of DMTRL (Yang & Hospedales, 2017). To be fair, all the methods are adopted with same network architecture. We repeat the experiments five times and record the average classification accuracy. As for the sharing, we tensorize the layer-wise weight into a $D$th-order tensor, whose $D$ modes have roughly the same dimensionality, such that the cores are approximately equal if we assume the same TR-ranks. Therefore, we may measure the faction of sharing by the number of cores $c$, which is needed to tune via cross validation. The search space of this hyper-parameter grows rapidly as number of the layers increase. In practice, we can mitigate this issue a lot by following a useful guidance that this number tends to decrease as the layers increase. Another solution is to apply a greedy search on $c$ layer by layer to effectively reduce the searching space. At last, we employ a similar trick introduced in (Yang & Hospedales, 2017) to specify the TR-ranks $R$ (or number $D$).

We conduct our experiments on following datasets: **MNIST** LeCun et al. (1998) contains handwritten digits from zero to nine. For this dataset, the task A is to classify the odd digits and the task B is to classify the even ones. **CIFAR-10** (Krizhevsky & Hinton, 2009) contains $60,000$ colour images of size $32 \times 32$ from 10 object classes. We assign 10 classes into 3 tasks, in which task A relates to non-animals; task B comprises 4 animal classes including 'cat', 'dog', 'deer' and 'horse'; tasks C contains the remaining 2 classes. **Omniglot** (Lake et al., 2015) consists of 1623 unique characters from 50 alphabets. There are only 20 examples for every character, drawn by a different person at resolution of $105 \times 105$. We divide the whole alphabets into five tasks (A to E), each of which links to the alphabets from 10 different languages. In the **Omniglot-MNIST** multi-dataset setting, the task A is assigned to classify the first 10 alphabets, while the task B is to recognize 10 digits. Due to the paper limit, please refer to the appendix for architectures and more experimental results.

### 5.1 VALIDATION ON SHARING PATTERNS AND MODEL COMPACTNESS

In order to see how sharing styles affect our performance, we examine various patterns from three representative categories, as shown in Figure 4. For instance, the patterns in 'bottom-heavy' category mean more parameters are shared at the bottom layers than the top layers, while 'top-heavy' indicates the opposite style. The validation is conducted on MNIST using MLP with three tensorized hidden layers, each of which is encoded using 4 TR-cores. The pattern '014', for example, means the $c$ are 0, 1 and 4 from lower to higher layers, respectively. We gauge the transferability between tasks with unbalanced training samples by the averaged accuracy on the small-sample tasks. Clearly, the 'bottom-heavy' patterns achieve significantly better results than those from the other two categories. The pattern '420' makes a lot sense and obviously outperforms the pattern '044' in Figure 4, since

| Samples A vs B | STL | | MRN | | Tucker | | DMTRL-TT | | Ours-410 | | Ours-420 | |
|---|---|---|---|---|---|---|---|---|---|---|---|---|
| | A | B | A | B | A | B | A | B | A | B | A | B |
| 1800 vs 1800 | 96.8 | 96.9 | 96.4 | 96.6 | 95.2 | 96.2 | 96.2 | 96.7 | **97.5** | **97.7** | 97.4 | 97.6 |
| 1800 vs 100 | 96.8 | 88.1 | 96.5 | 88.6 | 95.2 | 85.5 | 96.1 | 86.3 | **97.6** | **90.2** | 97.5 | 89.9 |
| 100 vs 1800 | 88.0 | 96.9 | 89.3 | 96.5 | 85.4 | 96.6 | 87.1 | 96.6 | 90.1 | 97.5 | **90.3** | **97.6** |
| 100 vs 100 | 88.0 | 88.1 | 88.2 | 88.4 | 84.3 | 84.8 | 86.8 | 86.0 | 88.7 | **89.6** | **89.2** | 89.5 |

Table 1: Performance comparison of STL, MRN, DMTRL and our TRMTL on MNIST.

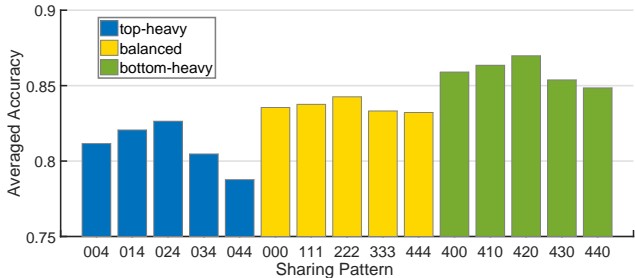

Figure 4: The averaged accuracy of two tasks involved with 50 samples. The training samples for 'task A vs task B' are '1800 vs 50' and '50 vs 1800'.

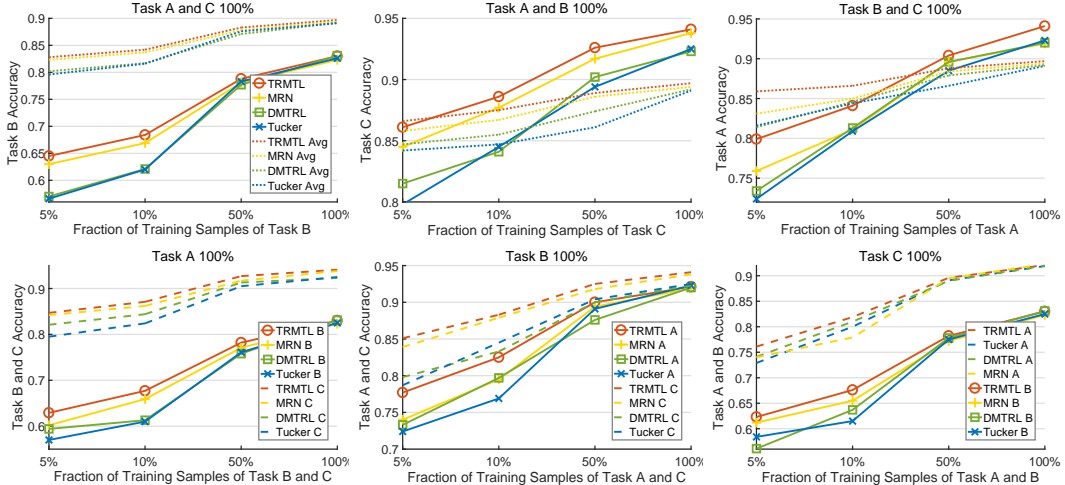

Figure 5: Performance comparison of MRN, DMTRL-Tucker, DMTRL-TT and our TRMTL-4431 on CIFAR-10 with different fractions of training data. Top row: 100% data for two of the three tasks, and show the accuracy for the other one task (in solid lines) as well as the averaged accuracy of all three tasks (in dotted lines). Bottom row: 100% data for one of the three tasks, and show the accuracies for the other two tasks (in dashed and solid lines).

'044' overlaps all weights at the top layers but shares nothing at the bottom layer. Within each category, TRMTL is robust to small perturbation of $c$ for pattern selection. For example, also in Table 1, both '410' and '420' patterns obtain similarly good performance. As for the model complexity, STL and MRN have enormous 6060K and 3096K parameters, since they share weights in the original space. DMTRL-Tucker and TT (1800 vs 1800) are parameterized by a large number of parameters of 1194K and 1522K. With TRMTL, this number is significantly down to 13K. The huge reduction is mainly due to the tensorization and the resulting more sparser TRL with overall lower ranks.

## 5.2 Results on tasks with insufficient samples

In this section, we like to verify the effectiveness of different models in transferring the useful knowledge from data-abundant task to data-scarcity task. To this end, we first test on CIFAR dataset using CNN with settings where each task may have insufficient training samples like 5%, 10% or 50%. Figure 5 illusrates how the accuracies of one task (two tasks) vary with sample fractions, given the remaining two tasks (one task) get access to the full data. We observe the trends in which the accuracies of our model exceed the other competitors by a relatively large margin (shown in solid lines), in the cases of limited training samples, e.g., 5% or 10%. In the mean time, the advantage of our TRMTL is still significant in terms of the averaged accuracies of three tasks (shown in dash lines), which implies the data-scarcity task has little bad influence on the data-abundant tasks. Our second test is carried out on the Omniglot with CNN architecture. We now test a more challenging case, where only 1 task (task C) has sufficient samples while the samples of the other 4 tasks (task A, B, D and E) are limited. Figure 6 demonstrates the amount of the accuracy changes for each task,

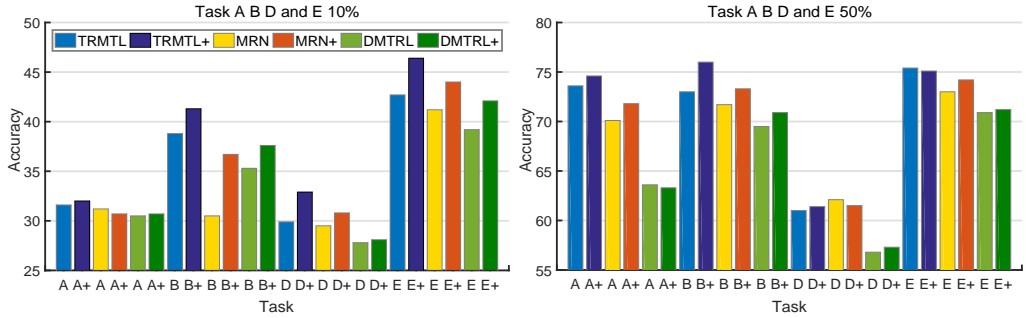

Figure 6: The results of accuracy changes of tasks A, B, D and E, when the faction of the data for training for task C is increased from 10% to 90%. '+' corresponds to the results after the samples augmentation of task C. Left (Right): 10% (50%) data for training for task A, B, D and E.

both with and without the aid of the data-rich task. We observe our TRMTL is able to make the most of the useful knowledge from task C and cause the accuracy to increase for most of the time. Particularly, the gap of the accuracy enhancement is more obvious for the case of 10% data.

## 5.3 RESULTS ON TASKS WITH HETEROGENEOUS INPUT DIMENSIONALITY

We next show the advantage of our method in handling tasks with heterogeneous inputs. In this test, the tasks are assigned to input images with different spatial sizes or distinct channels (i.e. RGB or grayscale). In order to apply DMTRL, one has to first convert the heterogeneous inputs into equal-sized features using *one hidden layer with totally unshared weights*, so that the weights in following layers can be stacked up and factorized. To better show the influence of heterogeneous inputs on the competitors, we adopt MLP with 4 hidden layers. For a good pattern of our TRMTL, such as '5410', the first hidden layer of each task is encoded into 6 TR-cores, 5 of which can be shared. As recorded in Table 2, DMTRL based methods behave significantly worse than our TRMTL by a very large margin. The poor performance of DMTRL is induced by fact that lowest features from related tasks cannot be shared at all because of the heterogeneous input dimensionality.

| Model | RGB A | Gray B | Gray C | Gray A | RGB B | Gray C | 16×16 A | 32×32 B | 16×32 C | 32×32 A | 16×32 B | 16×16 C |
|---|---|---|---|---|---|---|---|---|---|---|---|---|
| STL | 74.0 | 56.8 | 77.3 | 68.4 | 62.3 | 77.3 | 73.9 | 62.3 | 82.2 | 74.0 | 62.0 | 83.1 |
| DMTRL-Tucker | 72.8 | 55.2 | 76.6 | 66.6 | 61.6 | 77.2 | 72.9 | 60.9 | 82.1 | 73.1 | 61.2 | 82.5 |
| DMTRL-TT | 73.1 | 54.1 | 77.2 | 66.2 | 61.5 | 77.4 | 72.3 | 61.9 | 82.5 | 73.1 | 62.2 | 82.2 |
| TRMTL-5410 | **79.4** | **59.3** | **82.9** | **73.5** | **64.9** | **83.5** | **74.8** | **63.2** | **86.8** | **74.9** | **62.8** | **86.6** |

Table 2: The results of heterogenous input dimensionality on CIFAR-10. Left columns: each task associates with RGB or grayscale image. Right columns: tasks with images of different spatial sizes.

## 5.4 RESULTS ON TASKS FROM MULTIPLE DATASETS

Our TRMTL also finds its usefulness when applied to multiple datasets, where the tasks are loosely related. We verify this through recognizing character symbols (task A on Omniglot) and handwritten digits (task B on MNIST) at the same time. Task A is much harder than task B, as each character in task A has much fewer training samples. TRMTL is established using three hidden layers with 5 cores at each layer. Task A and B are partially shared by 2 cores at the first layer. To apply DMTRL, we use a similar strategy as previous section. As expected, TRMTL outperforms other methods and TRMTL-211 significantly improves task A by 4.2%, 4.9% and 4.7% w.r.t. STL, whereas both DMTRL-Tucker and TT fail in the Omniglot task with poor accuracies.

| Omniglot A vs MNIST B | STL | | DMTRL-Tucker | | DMTRL-TT | | TRMTL-200 | | TRMTL-211 | |
|---|---|---|---|---|---|---|---|---|---|---|
| | A | B | A | B | A | B | A | B | A | B |
| 50% vs 100% | 55.0 | 98.1 | 47.3 | **98.5** | 50.0 | 98.4 | 58.4 | 98.3 | **59.7** | 98.3 |
| 70% vs 100% | 60.5 | 98.1 | 46.1 | 98.3 | 50.9 | **98.6** | 62.9 | 98.3 | **65.4** | 98.3 |
| 100% vs 100% | 63.3 | 98.1 | 50.7 | **98.5** | 52.3 | **98.5** | 66.8 | 98.3 | **67.5** | 98.3 |

Table 3: The results of multi-dataset tasks on Omniglot (task A) and MNIST (task B).

## 6 CONCLUSION

In this paper, we have introduced a novel knowledge sharing mechanism for connecting task-specific models in deep MTL, namely TRMTL. The proposed approach models each task separately in the form of TR representation using a sequence latent cores. Next, TRMTL shares the common knowledge by ting any subset of layer-wise TR cores among all tasks, leaving the rest TR cores for private knowledge. TRMTL is highly compact yet super flexible to learn both task-specific and task-invariant features. TRMTL is empirically verified on various datasets and achieves the state-of-the-art results in helping the individual tasks to improve their overall performance.

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

| Samples A vs B vs C | Task | STL | MRN | DMTRL | Ours-4444 | Ours-4431 | Ours-4421 |
|---|---|---|---|---|---|---|---|
| 100% vs 100% vs 100% | A | 91.4 | 91.8 | **92.2** | 90.6 | 92.1 | 92.2 |
| | B | 80.9 | 82.3 | 82.3 | 81.6 | **83.0** | 82.6 |
| | C | 91.8 | 93.9 | 92.3 | 93.4 | **94.1** | 93.9 |
| | Average | 88.0 | 89.3 | 88.9 | 88.5 | **89.8** | 89.6 |
| 5% vs 5% vs 5% | A | 72.7 | 72.9 | 73.7 | 72.4 | 74.4 | **74.7** |
| | B | 57.0 | 60.3 | 55.5 | 59.0 | 61.3 | **61.5** |
| | C | 80.6 | 82.7 | 79.5 | 82.7 | 82.9 | **84.4** |
| | Average | 70.1 | 72.0 | 69.6 | 71.4 | 72.9 | **73.5** |
| 5% vs 5% vs 100% | A | 72.7 | 73.3 | 74.2 | 73.6 | 76.1 | **76.4** |
| | B | 57.0 | 61.2 | 56.3 | 60.2 | 62.3 | **63.0** |
| | C | 91.8 | 92.1 | 91.5 | 91.8 | **93.1** | 93.0 |
| | Average | 73.8 | 75.5 | 74.0 | 75.2 | 77.2 | **77.4** |
| 5% vs 100% vs 100% | A | 72.7 | 75.9 | 74.3 | 76.9 | **79.9** | 79.8 |
| | B | 80.9 | 80.2 | 79.7 | 79.5 | **81.2** | .81.1 |
| | C | 91.8 | 93.3 | 92.1 | 92.7 | **93.9** | **93.9** |
| | Average | 81.8 | 83.1 | 82.0 | 83.0 | **85.0** | 84.9 |

Table 4: Performance comparison of STL, MRN, DMTRL and our TRMTL on CIFAR-10 with unbalanced training samples, e.g., '5% vs 5% vs 5%' means 5% of training samples are available for the respective task A, task B and task C. TR-ranks $R = 10$ for TRMTL.

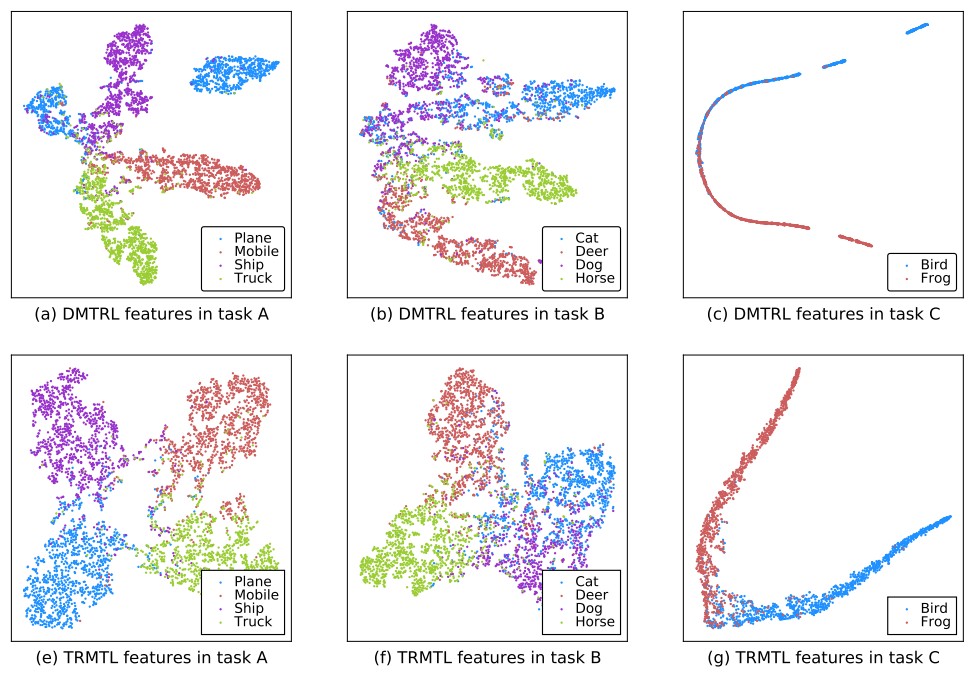

Figure 7: Features visualization of 2000 CIFAR-10 images. Tasks A, B and C correspond to three categories in CIFAR-10, i.e., non-animals, animals with bird and frog excluded, bird and frog. Top row: DMTRL features. Bottom row: our features.

## 6.1 MORE RESULTS ON TASKS WITH INSUFFICIENT SAMPLES ON CIFAR-10

In this section, we conduct more experiments on CIFAR-10 dataset. We adopt the following architecture: $(3 \times 64\,C3) - (64 \times 128\,C3) - (128 \times 256\,C3) - (256 \times 512\,C3) - (8192 \times 1024\,FC) - (1024 \times 512\,FC) - (512 \times 10\,FC)$, where $C3$ stands for a $3 \times 3$ convolutional layer. We employ TRL

| Samples A vs B vs C vs D vs E | Task | STL | MRN | DMTRL | Tucker | Ours-432 |
|---|---|---|---|---|---|---|
| | A | 30.4 | 31.2 | 30.5 | 28.9 | **31.6** |
| | B | 32.4 | 35.4 | 35.3 | 32.9 | **38.9** |
| 10% vs 10% vs 10% vs 10% vs 10% | C | 47.5 | 47.8 | 44.1 | 47.9 | **48.2** |
| | D | 29.2 | 29.5 | 27.8 | 28.4 | **29.9** |
| | E | 40.5 | 41.2 | 38.7 | **43.0** | 42.7 |
| | Average | 36.0 | 37.0 | 35.3 | 36.2 | **38.3** |
| | A | 61.1 | 70.1 | 63.6 | 59.0 | **73.6** |
| | B | 66.4 | 71.7 | 69.5 | 67.3 | **73.0** |
| 50% vs 50% vs 50% vs 50% vs 50% | C | 73.1 | 77.8 | 75.3 | 70.9 | **80.5** |
| | D | 55.8 | **62.1** | 56.8 | 55.8 | 61.0 |
| | E | 68.8 | 73.0 | 70.9 | 71.0 | **75.4** |
| | Average | 65.0 | 70.9 | 67.2 | 64.8 | **72.7** |
| | A | 72.2 | 78.6 | 74.0 | 75.5 | **80.5** |
| | B | 75.4 | **80.7** | 77.9 | 76.4 | 79.5 |
| 90% vs 90% vs 90% vs 90% vs 90% | C | 82.7 | 86.5 | 81.7 | 82.5 | **88.7** |
| | D | 60.5 | 69.7 | 65.3 | 62.7 | **72.2** |
| | E | 74.9 | **82.1** | 76.7 | 75.4 | 80.7 |
| | Average | 73.1 | 79.5 | 75.1 | 74.5 | **80.3** |

Table 5: Performance comparison of STL, MRN, DMTRL and our TRMTL on Omniglot with different fractions of training samples.

on the last two CNN layers and first two FC layers, in which the most of the parameters concentrate, yielding 4 TR-cores per layer.

We show more results on the effectiveness of different models when transferring the useful knowledge from data-abundant task to data-scarcity task. For this purpose, we begin with the test cases where all of task have insufficient training samples, e.g., '5% vs 5% vs 5%'. After that, we compare the precision improvement of the individual task(s) when the other task(s) is (are) equipped with the whole training data. Table 4 records the results of our two best patterns ('4431' and '4421'), as well as the one with 'bad' pattern '4444'. Clearly, TRMTL ('4431' and '4421') outperforms other methods in nearly all the cases. As for task A, for instance, the precision of TRMTL-4431 is increased by 1.7% when the data of the task C becomes 100%. Even more, such enhancement further grows up to 5.5% in the situation that both task B and C's training samples are fully available. This is in contrast to MRN whose precision improvements are merely 0.4% and 3.0% in the corresponding scenarios. Again, the performance of TRMTL-4431 is superior to that of TRMTL-4444, indicating sharing all nodes like '4444' is not a desirable style.

It is also interesting to get an idea on what our model has learned via the visualization of the high level features. Figure 7 illustrates the task-specific features of our TRMTL (and DMTRL) using t-SNE for the dimensionality reduction. We can see a clear pattern of the clustered features produced by our model that are separated for different classes, which could be more beneficial the downstream classification tasks.

## 6.2 MORE RESULTS ON TASKS WITH INSUFFICIENT SAMPLES ON OMNIGLOT

For this dataset, we adopt a similar architecture as in the previous experiment for CNN as

$$(1 \times 8\ C3) - (8 \times 16\ C3) - (16 \times 32\ C3) - (23,328 \times 256\ FC) - (256 \times 50\ FC),$$

where the last two convolution layers and first fully connected layer are represented using TRL with the input/output feature modes of TR-cores being $\{2,2,2\}$, $\{4,2,2\}$, and $\{2,2,2,2\}$, $\{4,4,2,2\}$, and $\{18,12,12,9\}$, $\{4,4,4,4\}$. The best sharing pattern of our model is '432', which is selected by CV. Table 5 summarizes the performance of the compared methods when the distinct fractions of data are used as training data. Our TRMTL obtains the best overall performance in both data-rich and data-scarcity situations.

## 6.3 MORE RESULTS ON TASKS WITH HETEROGENEOUS INPUT DIMENSIONALITY

Table 6 records the complete results on tasks with heterogeneous input dimensionality.

| Model | RGB A | Gray B | Gray C | Gray A | RGB B | Gray C | Gray A | Gray B | RGB C |
|---|---|---|---|---|---|---|---|---|---|
| STL | 74.0 | 56.8 | 77.3 | 68.4 | 62.3 | 77.3 | 68.4 | 56.8 | 83.2 |
| DMTRL-Tucker | 72.8 | 55.2 | 76.6 | 66.6 | 61.6 | 77.2 | 66.3 | 55.4 | 82.6 |
| DMTRL-TT | 73.1 | 54.1 | 77.2 | 66.2 | 61.5 | 77.4 | 66.7 | 54.8 | 81.7 |
| TRMTL-5410 | **79.4** | **59.3** | **82.9** | **73.5** | **64.9** | **83.5** | **74.4** | **59.4** | **88.9** |

| Model | 16×16 A | 32×32 B | 16×32 C | 32×32 A | 16×32 B | 16×16 C | 16×32 A | 16×16 B | 32×32 C |
|---|---|---|---|---|---|---|---|---|---|
| STL | 73.9 | 62.3 | 82.2 | 74.0 | 62.0 | 83.1 | 74.3 | **62.4** | 82.2 |
| DMTRL-Tucker | 72.9 | 60.9 | 82.1 | 73.1 | 61.2 | 82.5 | 72.6 | 61.2 | 82.6 |
| DMTRL-TT | 72.3 | 61.9 | 82.5 | 73.1 | 62.2 | 82.2 | 73.4 | 61.5 | 82.8 |
| TRMTL-5410 | **74.8** | **63.2** | **86.8** | **74.9** | **62.8** | **86.6** | **75.2** | **62.4** | **86.7** |

Table 6: The results of heterogenous input dimensionality on CIFAR-10. Top: each task associates with RGB or grayscale image. Bottom: each task has input images of different spatial sizes.

| MNIST | | Task A (Odd) | Task B (Even) |
|---|---|---|---|
| Layer 1 | input modes | [7, 7, 4, 4] | [7, 7, 4, 4] |
| | output modes | [6, 6, 6, 6] | [6, 6, 6, 6] |
| Layer 2 | input modes | [6, 6, 6, 6] | [6, 6, 6, 6] |
| | output modes | [6, 6, 6, 6] | [6, 6, 6, 6] |
| Layer 3 | input modes | [6, 6, 6, 6] | [6, 6, 6, 6] |
| | output modes | [4, 4, 4, 4] | [4, 4, 4, 4] |
| Layer 4 | input modes | [256] | [256] |
| | output modes | [10] | [10] |

Table 7: Specification of network architecture and factorized TRL representation of the experiments for the validation of sharing pattern and model compactness on MNIST dataset.

| Omniglot | Kernel/Weight | | Task A | | Task B | |
|---|---|---|---|---|---|---|
| Layer 1 | input modes | window size | [1] | [3, 3] | [1] | [3, 3] |
| | output modes | | [8] | | [8] | |
| Layer 2 | input modes | window size | [2, 2, 2] | [3, 3] | [2, 2, 2] | [3, 3] |
| | output modes | | [4, 2, 2] | | [4, 2, 2] | |
| Layer 3 | input modes | window size | [2, 2, 2, 2] | [3, 3] | [2, 2, 2, 2] | [3, 3] |
| | output modes | | [4, 2, 2, 2] | | [4, 2, 2, 2] | |
| Layer 4 | input modes | | [18, 16, 9, 9] | | [18, 16, 9, 9] | |
| | output modes | | [4, 4, 4, 4] | | [4, 4, 4, 4] | |
| Layer 5 | input modes | | [256] | | [256] | |
| | output modes | | [10] | | [10] | |

Table 8: Specification of network architecture and factorized TRL representation of the experiments for the insufficient sample tasks on Omniglot dataset.

| Heterogenous Spatial Sizes | | Task A (RGB) | Task B (Gray) | Task C (Gray) |
|---|---|---|---|---|
| Layer 1 | input modes | [4, 4, 4, 4, 4, 3] | [4, 4, 4, 4, 4, 1] | [4, 4, 4, 4, 4, 1] |
| | output modes | [4, 4, 4, 4, 4, 6] | [4, 4, 4, 4, 4, 6] | [4, 4, 4, 4, 4, 6] |
| Layer 2 | input modes | [8, 8, 6, 4, 4] | [8, 8, 6, 4, 4] | [8, 8, 6, 4, 4] |
| | output modes | [8, 8, 6, 4, 8] | [8, 8, 6, 4, 8] | [8, 8, 6, 4, 8] |
| Layer 3 | input modes | [8, 8, 6, 8, 4] | [8, 8, 6, 8, 4] | [8, 8, 6, 8, 4] |
| | output modes | [8, 8, 6, 8, 8] | [8, 8, 6, 8, 8] | [8, 8, 6, 8, 8] |
| Layer 4 | input modes | [8, 8, 6, 8, 8] | [8, 8, 6, 8, 8] | [8, 8, 6, 8, 8] |
| | output modes | [3, 3, 3, 3, 3] | [3, 3, 3, 3, 3] | [3, 3, 3, 3, 3] |
| Layer 5 | input modes | [243] | [243] | [243] |
| | output modes | [10] | [10] | [10] |

Table 9: Specification of network architecture and factorized TRL representation of the experiments on heterogenous inputs with distinct spatial sizes for CIFAR-10.

| RGB and Grayscale | | Task A (16×32) | Task B (16×16) | Task C (32×32) |
|---|---|---|---|---|
| Layer 1 | input modes | [4, 4, 4, 4, 3, 2] | [4, 4, 4, 4, 3, 1] | [4, 4, 4, 4, 3, 4] |
| | output modes | [4, 4, 4, 4, 3, 8] | [4, 4, 4, 4, 3, 8] | [4, 4, 4, 4, 3, 8] |
| Layer 2 | input modes | [8, 8, 6, 4, 4] | [8, 8, 6, 4, 4] | [8, 8, 6, 4, 4] |
| | output modes | [8, 8, 6, 4, 8] | [8, 8, 6, 4, 8] | [8, 8, 6, 4, 8] |
| Layer 3 | input modes | [8, 8, 6, 8, 4] | [8, 8, 6, 8, 4] | [8, 8, 6, 8, 4] |
| | output modes | [8, 8, 6, 8, 8] | [8, 8, 6, 8, 8] | [8, 8, 6, 8, 8] |
| Layer 4 | input modes | [8, 8, 6, 8, 8] | [8, 8, 6, 8, 8] | [8, 8, 6, 8, 8] |
| | output modes | [3, 3, 3, 3, 3] | [3, 3, 3, 3, 3] | [3, 3, 3, 3, 3] |
| Layer 5 | input modes | [243] | [243] | [243] |
| | output modes | [10] | [10] | [10] |

Table 10: Specification of network architecture and factorized TRL representation of the experiments on heterogenous inputs with distinct channels (RGB and grayscale image) for CIFAR-10.

| Omniglot-MNIST | | Task A (105×105) | Task B (28×28) |
|---|---|---|---|
| Layer 1 | input modes | [7, 7, 5, 5, 3, 3] | [7, 7, 4, 4] |
| | output modes | [7, 7, 5, 5, 3, 3] | [7, 7, 5, 5] |
| Layer 2 | input modes | [7, 7, 5, 5, 3, 3] | [7, 7, 4, 4] |
| | output modes | [7, 7, 5, 5, 3, 3] | [7, 7, 5, 5] |
| Layer 3 | input modes | [7, 7, 5, 5, 3, 3] | [7, 7, 4, 4] |
| | output modes | [2, 2, 2, 2, 2, 2] | [2, 2, 2, 2] |
| Layer 4 | input modes | [64] | [16] |
| | output modes | [10] | [10] |

Table 11: Specification of network architecture and factorized TRL representation of the experiments for multi-dataset task on Omiglot-MNIST.

