# OpenReview forum: "TENSOR RING NETS ADAPTED DEEP MULTI-TASK LEARNING"
_ICLR.cc/2019/Conference_

### Official Review · AnonReviewer3 · 2018-11-02
**Tensor-based soft-sharing MTL, upgraded with TR-decomposition. Maybe practically useful to enhance. But not clearly enough written or evaluated.**

**Rating:** 4
**Confidence:** 4

**Review:**

Summary:  This paper studies deep multi-task learning. Prior papers have studied various knowledge sharing approaches for deep multi-task learning including hard and soft sharing schemes. And some soft sharing schemes have used tensor decompositions (including TT, and Tucker). This paper fuses this line of work with the recently proposed Tensor-Ring decomposition in order to obtain Tensor Ring (TR)-based soft sharing for multi-task learning. The results show some improvement over prior deep MTL methods based on other tensor factorisation methods.

Strengths:
+ Nice extension of existing line of work tensor-factorisation based MTL.
+ More flexibility for controlling shared/unshared portions of weights compared to DMTRL.
+ Improves on previous methods results.
+ Experiments evaluate how MTL methods relate with various amounts of training data on each task.

Weaknesses:
- Novelty/significance is limited.
- Writing. Many things are not clearly and intuitively explained. Some claims are not adequately justified.
- Introduces more hyper parameters to tune.
- Results may rely on hyper parameter tuning.

Comments:
1. Novelty.  Existing studies already established the template of different tensor factorisation methods (TT, Tucker) being possible to plug into deep networks for different kinds of soft-sharing MTL. Meanwhile, TR decomposition is taken off the shelf; (and as it’s been applied for compression before, this is not the first time TR decomposition has been used in a CNN context either). Therefore this is an A+B paper and a high bar should be met for the additional analysis, insight, or performance improvements that should provided.
2. Lots of writing issues:
2.1 Many things are not explained transparently enough at best (or major over-claim at worst). For example:
2.1.1 Paper claims the benefit that each task can have its own I/O dimensionality. However if TR-decomp is “circularly connected” TT-decomp (Fig 1), then this seems not to happen automatically. So it should be unpacked more clearly how this is achieved.
2.1.2 Paper claims favourable ability to use more private cores than TT, where only one core is private. However circular TT would also seem to have one private core by default (the core with a task axis). So I suspect something else is going on, but this is completely unclear and should be explained more transparently. Furthermore it should be justified if whatever modifications do enable these properties are definitely a unique property of TR-decomp, or could also be applied to TT-decomp.
2.1.3 Statement “TRMTL generalizes to allow the layer-wise weight to be represented by a relatively lager number of latent cores” unclear: generalises what? larger number of cores than what? Than TT? The previous presentation suggests TT and TR should have same number of cores.
2.1.4 Statements like “TR enjoys the property of circular dimensional permutation invariance” are made without any explanation about what is the implication of this for neural networks and multi-task learning.
2.2 Many claims are inaccurate or not adequately backed up by theory or experiment. EG: (i) Paper claims to include DMTRL as a special case. But it only subsumes DMTRL-TT, not DMTRL-Tucker. Because TR-decomp does not include Tucker-decomp as an exact special case.  (ii) Sentences “TR-ranks are usually smaller than TT-ranks” are assertions without verification.
2.3 Sentences are taken verbatim from other papers, plagiarism. For example: “TR model is more flexible than TT, because TR-ranks can be equally distributed in the cores, but TT-ranks have a relatively fixed pattern”  is verbatim from Zhao’16 TR-decomp paper.
3. Hyperparameters: This paper apparently gains some practical benefit due to the notion of shared/unshared cores. However, this also introduces  additional hyper parameters (E.g., each layers private proportion “c”) to tune besides the ranks. Unlike the rank that can be pre-estimated by reconstruction error, this one seems to require tuning by cross-validation. This is not scalable.
4. Hyperparameters+Tuning: Hyperparameters Private proportion, “sharing pattern”, IO dimension seem to be tuned by accuracy.( “We test different sharing patterns and report the ones with the best accuracies”). This is even less scalable, and additional tuning makes it unsurprising it surpasses other models performance.
5. Insight & Analysis. All the core selection & public/private core selection are treated as black box optimisation. No insight is given about what turns out to be useful to share or not, and how consistent this is, etc.

---

> ### Author Response · Authors · 2018-11-26
> **Improvement in writing, motivation and evaluation in the revised paper, to clarify the misunderstandings of the proposed method (feedback part1)**
>
> We would like to thank the Reviewer for the insightful and constructive questions and comments.
>
> 1. ''Novelty. Existing studies already established the template of different tensor …’'
> We respectfully disagree with the Reviewer on this comment. Firstly, to the best of our knowledge, DMTRL is the only framework integrates tensor factorization (TT, Tucker) into deep MTL but with a highly restricted soft-sharing mechanism. Secondly, there is no any good solution or template to plug tensor factorization into different kinds of soft-sharing deep MTL. Most importantly, three major challenges posed by deep MTL remain largely unsolved, which includes: 1) the ineffectiveness of knowledge sharing mechanism; 2) the inefficiency of size of parameters for deep MTL model; 3) the inflexibility of network architectures to handle heterogeneous features and inputs. None of existing deep MTL models (especially DMTRL) can well handle all above difficulties.
>
> Our incorporation of TR into deep MTL serves for special purposes:  1) flexible/finer granularity of knowledge sharing (via tensorization); 2) higher compression ratio of deep MTL model sizes;  3) a generalized expressivity power (of TT);  4) better performance.
>
> This is why we come up TRMTL, the main novelty/significance of which is that we propose a totally new generalized, highly flexible and latent-subspace knowledge sharing framework that can effectively address all these challenges. It should be noted that by simply combining DMTRL template (A) and TR-format (B), it cannot achieve such goals.
>
> 2. ''2.1.1 Paper claims the benefit that each task can have its own I/O dimensionality …’'
> We agree that the explanation should be more clear. The benefit/flexibility of our TRMTL for heterogenous inputs mainly comes from our proposed architecture and sharing mechanisms, not due to the TR format. In fact, our framework can accommodate more generalized tensor network formats including TT or TR as a special case.
>
> 3. ''2.1.2 Paper claims favourable ability to use more private cores than TT …’'
> The Reviewer has a misunderstanding about our framework, especially the relations between DMTRL and our TRMTL (Please refer to Section 2 in the revised paper for details). DMTRL has to stack up all the equal-sized weights from different tasks, and hence has the ‘task’ axis.  In contrast, our TRMTL decomposes each task's own weight individually, then any subset of the cores can be shared among tasks. Therefore, it is not correct to say that TRMTL shares one core at default because our method is even possible to sharing zero core, and there is no such ’task’ axis involved.
>
> Our framework is so generalized that TT or other tensor network can also be subsumed into our architecture. However, using TR instead of TT does bring benefits which are very preferable by our framework, e.g., higher compression ratio (lower overall-ranks via tensorization) of TR allows for the sharing among more compact (smaller-sized) cores, which has a big impact on parameter complexity for deep MTL models.
>
> 4. ''2.1.3 Statement “TRMTL generalizes to allow the layer-wise weight … ‘’
> The sentence should have been more clear. We meant to say that our TRMTL framework generalizes DMTRL in terms of 4 major aspects. (Please see Section 2 in revised version), and one aspect of generalization is that TRMTL firstly tensorizes the weight into a much higher order weight tensor before factorizing it (In DMTRL paper, they did not employ tensorization). By doing so, the weight can be factorized into more cores than of just 3 cores for MLP (or 5 cores for CNN ) in DMTRL.
>
> 5. “2.1.4 Statements like “TR enjoys the property of circular dimensional permutation …"
> We present the 'circular dimensional permutation invariance’ property only in tensor preliminaries (Section 3). This property is one advantage of TR over TT,  and we introduce this only as a background knowledge, which is not related to our current work.

---

> ### Author Response · Authors · 2018-11-26
> **Improvement in writing, motivation and evaluation in the revised paper, to clarify the misunderstandings of the proposed method (feedback part2)**
>
> We would like to thank the Reviewer for the insightful and constructive questions and comments.
>
> 6. “2.2 Many claims are inaccurate or not adequately backed up  …”
> We agree that the explanation should have been formulated more accurately. However, we like to emphasize that there are four major types of generalizations from DMTRL to our TRMTL. (Please see Section 2). Our TRMTL indeed generalizes or subsumes DMTRL-Tucker in terms of the first three types/aspects, no matter what tensor format (TT or Tucker) DMTRL may use.
>
> “TR-ranks are usually smaller than TT-ranks’’ is presented in tensor preliminaries section just as background knowledge. Besides, this property has already been verified in papers [Zhao16,17], which we added the citation. Verifying this property is not the focus and the objective of our work.
>
> 7. ''2.3 Sentences are taken verbatim from other papers, plagiarism. For example: “TR model … ‘'
> We respectfully disagree with the Reviewer that it seems several 'Sentences are taken verbatim from other papers’, and this is clearly not the case.
> Only in the sentence 'TR model is more flexible than … ', we use the original expression for the purpose of exactly expressing the meaning that [Zhao16] wants to express in their paper. This sentence is written in the tensor preliminaries section and we added [Zhao16] as citation in that section. However, it would be better if we should have used our own words.
>
> 8. “3. Hyperparameters: This paper apparently gains some practical benefit …”
> We respectfully disagree with the Reviewer. Our TRMTL can effectively deal with all major challenges of current deep MTL. The benefits of TRMTL can totally compensate for the downside of the introduction of the parameter of ‘c’. Besides, tuning one or two hyper-parameters is quite common in deep learning research. In practical, by using some heuristics, e.g. choosing the right sharing styles such as ‘bottom-heavy', the searching space of ‘c' can be greatly reduced. We may also employ a greedy search on ‘c’ layer by layer, which is much easier for tuning ‘c’.
>
> 9. “4. Hyperparameters+Tuning: Hyperparameters Private proportion …"
> We disagree with the Reviewers that the better performance is because of the additional tuning. In our experiment, the i/o dimension of tensorization is pre-fixed, and the location/order in which the cores of each task are arranged is also pre-fixed. We even tried empirically fixing the TR-ranks and also got fairly good performance. Besides TR-ranks, only the sharing portion ‘c’ is left to tune. If we are allowed to tune all above hyper-parameters, (a.k.a all the potential flexibility of our method is fulfilled), our performance would be much better than we reported in the paper.
>
> Regarding “We test different ... with the best accuracies”, the writhing should have been more accurate, we meant that the ‘c’ is turned, and the best accuracies is reported w.r.t only to this ‘c’.
>
> 10. “5. Insight & Analysis. All the core selection & public/private core selection …"
> We respectfully disagree that the core selection are treated as black box optimization. Some insights about the cores selection are shown in our experiments. In MNIST experiment of previous version, Figure 3 demonstrates that the styles of sharing pattern have significant impact on the performance. Within different style categories, our model is very robust to ‘c’ for the pattern selection. For example, in Table 1, both ‘410’ and ‘420’ obtain the similarly good performance, which means small variation in ‘c’ does not affect a lot on performance, if the right style category is determined. Also In CIFAR experiment, in Table 2, both good patterns ‘4421’ and ‘4431’ belong to ’bottom-heavy’ category and achieve similarly good accuracies, but clearly outperform the bad pattern ‘4444’ which belongs to the ‘balanced’ style category. Some remarks/strategy on core selection are also provided in the revised version, please see Section 4.3 for details.

---

### Official Review · AnonReviewer2 · 2018-11-02
**Simple idea with interesting results**

**Rating:** 5
**Confidence:** 4

**Review:**

The novelty and experiments are somewhat limited. Thus I am lowering my score.
-----------------------------------------------------------------------------------------------------------------------------------------------------------------------

The authors proposed a variant of tensor ring formulation for multi-task learning. They achieved that by sharing some of the TT cores for learning "common task" while learning individual TT cores for each separate tasks.

Pros:
1) Overall nice but simple extension of TT/ TR framework
2) Nice set of experiments which have shown improvement over standard TT/ TR framework for MTL.

Cons (and suggestions):
1) As to my knowledge TT/ TR have not been used for MTL before, I wonder if someone wanted to the proposed method is the only way to achieve it, so in that sense, it's a very "simple" extension.
2) Though authors called something called "TRL", I think it is just an indexing scheme so essentially the same idea of TR.
3) I wonder why authors suddenly mentioned about convolution in the end of section 3.1., looks very out of the place discussion.
4) I suggest in Section 3.2., make the shareable cores not adjacent in Eq. (4) as they claimed.
5) The experiments are somewhat "````````simplistic" and I believe the power of this sharing should have experimented on Taskonomy data (https://arxiv.org/pdf/1804.08328.pdf). Right now, the experimental setup is very much simplistic, which is one of the main points the authors should address.
6) Can the authors comment on the number of parameters used?
7) I wonder if the author can show some RNN/ LSTM experiment because some of the datasets used like OMLIGLOT/ MNIST are too simple to count as an experiment. Challenge will be to see the performance in challenging MTL.
8) I believe the authors should comment on the choice of c and the location of the shareable cores.

---

> ### Author Response · Authors · 2018-11-25
> **TRMTL looks 'simple' but is essentially nontrivial,  a fairly nice solution to all three major challenges of deep MTL**
>
> We would like to thank the Reviewer for the insightful and constructive suggestions and comments.
> 1. ''As to my knowledge ... it's a very "simple” extension’’
> We thank the Reviewer for pointing out our main contribution that we propose a new generalized highly flexible latent-subspace based knowledge sharing mechanism.
> There are major challenges in deep MTL that remains largely unaddressed:  1) the ineffectiveness of knowledge sharing mechanism  2) the inefficiency of size of parameters for deep MTL model  3) the inflexibility of network architectures to handle heterogeneous features and inputs.
> Although DMTRL has used TT/Tucker factorization, their model is rather restricted and unable to solve these issues. None of existing deep MTL models can well handle these difficulties (please see the revised paper for details ). Our extension looks 'simple' but is essentially nontrivial, because by using the proposed architecture, we can provide a fairly nice solution to all three major challenges.
>
> 2. ''Though authors called ... is just an indexing scheme so essentially the same idea of TR.’'
> We agree with the Reviewer that TRL is based on and very similar to TR, but we respectfully disagree that TRL is simply an indexing scheme. The reasons are described as follows.
> Tensorization ('indexing scheme') in TRL plays a key role in our proposed sharing mechanism. Due to the tensorization, we can share the cores in a much finer granularity. Moreover, tensorization in TRL can lead to more compact tensor network representation (with lower ranks), and thus a higher compression ratio for the parameters. (please see section 5.1 in the revised version for the comparison of model complexity, where DMTRL without tensoriztion versus ours with tensorization)
>
> 3. ''I wonder why ... looks very out of the place discussion.’'
> We mention this because this is variant of TRL for CNN setting, since in our experiments, we share the filters/kernels (itself is a 4th order tensor H x W x U x V) in CNN instead of sharing weight matrices in MLP between tasks. The two sharing settings are essentially the same but with slightly different formulation.
>
> 4. ''I suggest ...make the shareable cores not adjacent in Eq. (4) as they claimed’’
>  We thank the Reviewer for the nice suggestion, which is much better to make sharable cores not adjacent, we have updated it in the received version.
>
> 5. ''The experiments are somewhat ''simplistic" …’'
> We agree with the Reviewer that more experiments should be done on larger dataset, e.g. Taskonomy data. The Taskonomy data is somewhat huge and we have some difficulty to get access to it during this rebuttal, but we will try to add it in the next version. However, in this revised version, we have conducted more experiments to further validate the merits of our TRMTL. In section 5.3, we have tested on tasks with heterogeneous input dimensions. In section 5.4, we applied our method to multiple datasets settings, where tasks could be loosely related.
>
> 6. ''Can the authors comment on the number of parameters used?’'
> We reports the number of parameters for different models in section 5.1 to demonstrate the compactness of our model. Overall, STL (6060K) has enormous parameters; MRN (3096K) have huge number of parameters, since they share weights in the original space. DMTRL Tucker (1194K) and TT (1522K) also have large parameters as models does not employ tensorization and just decomposes the stacked weight tensor of original order. In contrast, our TRMTL only uses 13K parameters, which is about 100 times fewer than DMTRL. Our model first tensorizes the weight into a much higher order tensor before factorizing it. By doing so, the weights can be represented into larger number of cores but with much lower ranks via TRL, which yields a highly compact model.
>
> 7. ''I wonder if the author can show some RNN … ‘'
> We thank the Reviewer for the RNN advice. It would be interesting to see how method works on tasks with sequence data. However, such an experiment is not trial and will take some time,  since all the current compared methods are only designed for MLP/CNN. We will try to add this in future version.
>
> 8. ''I believe the authors should comment on …’’
> In the revised version, some remarks or discussions are given in section 4.3. In current work, the location of shared cores are arranged in a left-to-right order, since there is natural intuition on the connection between cores and image resolution [Zhao et al 17], in which the first core mainly controls the small patches while the last core affects the large patches. In this experiment, we preferentially share the features from detailed scale to coarse scale (share the cores from left to right). By fixing that, we only use ‘c’ to control the fraction of sharing. In future work, we plan to automatically select shared core pairs with highest similarity between tasks.

---

> > ### Comment · AnonReviewer2 · 2018-12-03
> > **Response**
> >
> > Thank you for the response.
> >
> > After reading other reviews and the responses from authors, I still believe that the paper has some merit. Though the authors tried to justify the novelty (and they did a good job in explaining I must say), I still believe in the following cons:
> >
> > (1) Novelty is somewhat limited and incremental.
> > (2) Experimental validation should be performed on a large-scale dataset. To me, the entire ``" selling pitch" for TT/ TR based methods is the applicability in large-scale data. Reduction in parameter otherwise cannot be much appreciated.
> > (3) Hyperparameter sensitivity is a crucial issue. My limited experience in TT/ TR tells me that it is indeed very sensitive to "tuning of hyper-parameters". On top of that, this method introduces more hyper-parameters. So that's not good.
> >
> > Now that we are almost there to make the final decision, I am sorry but I have to reject the paper as after reading other reviews and some thinking, I can not see this paper in its current form as an ICLR paper. But I highly appreciate the clarification and encourage the authors to do some large-scale experiment and possibly submit to the next ML/ CV  venue.

---

> > > ### Author Response · Authors · 2018-12-04
> > > **The new perspective of our contribution and novelty**
> > >
> > > Thank you for your response. We provide totally different perspective of our contribution and novelty.  Please give us one more opportunity by reading the following explanation:
> > >
> > > 1 ''(1) Novelty is somewhat limited and incremental.''
> > > The most significant novelty of our paper is the information sharing mechanisim in deep multitask learning. A major challenging problem in all existing deep multitask learning methods is that they cannot handle heterogeneous dataset and neural network structure (i.e., different dimensions of inputs and all layers) corresponding to the different tasks.  Our method solves this problem by using tensor ring representation of model parameters and then information sharing on the latent space.  As a consequence, our method can also provide more refined control of how much information to be shared and much reduction of model complexity. Therefore, as compared to the existing methods, our method is more effective, efficient, and flexible.
> > >
> > >
> > > 2 ''Experimental validation should be performed on a large-scale dataset ...''
> > > The contribution of this paper is not applicability in large-scale data and reduction of parameters by using TT/TR.  We discovered another totally different advantages by using TT/TR in deep multi-task learning, which is the refined information sharing control over the latent space rather than the hard-sharing and soft-sharing mechanism. Then we can control how much information is shared between each task in more detailed scales.
> > >
> > >
> > > 3  ''Hyper-parameter sensitivity is a crucial issue.''
> > > In hard-sharing method,  one hyper-parameter is the number of shared layers. In soft-sharing method, one hyperparameter is the penalty weight of regularization or the rank of task mode factor. As compared to existing deep multi-task learning method, our method also has only one hyperparameter that is how many cores are shared. Therefore, our method does not introduce more hyper-parameters.
> > >
> > > TT/TR is sensitive to the hyper-parameters, because their methods try to find the optimal balance between compression and performance. However, in this paper, we do not focus on the compression ability. We try to achieve the optimal performance when using information sharing on latent space. Thus, we usually fixed the TT/TR ranks to have sufficient representation ability, for example it can be the upper bound of TT/TR ranks.

---

### Official Review · AnonReviewer1 · 2018-11-04
**Poorly organized, poorly motivated paper.**

**Rating:** 6
**Confidence:** 4

**Review:**

Summary: The authors propose tensor ring nets for multi-task learning

Cons: This is a poorly organized paper and poorly motivated.
This paper discusses relevant mathematics with no motivation in section 2, while the  prior work is in section 4. Seems backward.

Please reference the first papers to employ tensor decompositions for imaging.

M. A. O. Vasilescu, D. Terzopoulos, "Multilinear Analysis of Image Ensembles: TensorFaces,"  Proc. 7th European Conference on Computer Vision (ECCV'02), Copenhagen, Denmark, May, 2002, in Computer Vision -- ECCV 2002, Lecture Notes in Computer Science, Vol. 2350, A. Heyden et al. (Eds.), Springer-Verlag, Berlin, 2002, 447-460.

 M. A. O. Vasilescu, D. Terzopoulos, "Multilinear Subspace Analysis for Image Ensembles,'' Proc. Computer Vision and Pattern Recognition Conf. (CVPR '03), Vol.2, Madison, WI, June, 2003, 93-99.

M.A.O. Vasilescu, "Multilinear Projection for Face Recognition via Canonical Decomposition ",  In Proc. Face and Gesture Conf. (FG'11), 476-483.

---

> ### Author Response · Authors · 2018-11-25
> **improvement with a clear organization and a better motivation**
>
> We would like to thank the reviewer for the insightful and constructive suggestions and comments.
>
> 1. ''This paper discusses ..., while the prior work is in section 4, seems backwards.''
> We agree that the paper is not well organized. In our revised version, we have reorganized our paper by putting the prior work in Section 2. We have revised introduction part in Section 1 with a much clearer motivation for our work. The Section 3 gives a brief tensor background knowledge without mathematics. The proposed method with its mathematical formulation is given in Section 4.
> Please note that we have completely rewritten the introduction part (Section 1) in order to give a strong and clear motivation of the proposed method. Please refer to the revised paper for more details.
>
> 2. ''Please reference the first papers to employ tensor decompositions for imaging’'
> The mentioned papers are the very original and popular work that successfully apply tensor decomposition to imaging analysis and also CV. We have referenced these work in the revised version.

---

### Author Response · Authors · 2018-11-26
**Summary of the revision**

We thank all the reviewers for their insightful and valuable reviews. In the revised version, we have made several major revisions and improved our paper in the following aspects:

1. Motivation
We have rewritten and improved the introduction part (Section 1) so as to have a strong and clear motivation of the proposed method.

2. Related work
We have revised the related work part (Section 2) to further illustrate the essential difference between our method and DMTRL, so as to help to clarify some misunderstandings about our method.

3. Experiment evaluation
We have added the following extra experiments, in order to support the claims on the advantages/properties of our method:
1) experiment on tasks with heterogenous input dimensionality
2) experiment on tasks from multiple datasets
3) report the model complexity

4. Paper structure
We have reorganized our paper for a better paper structure. For example, we moved the related work part from Section 4 to Section 2; we optimized structure of the experiment section (Section 5) to better highlight the key properties of the proposed method.

5. Paper writing
We have fixed the typos, and improved the overall quality in writing. For example, in tensor preliminaries part (Section 3), we give a clearer presentation of the TR background knowledge, so as not to misunderstand between [Zhao16,17]'s contribution/focus and ours.

---

### Meta-Review · Area_Chair1 · 2018-12-15
**Numerous concerns.**

**Confidence:** 5
**Recommendation:** Reject

**Metareview:**

AR1 is concerned about the poor organisation of this paper.  AR2 is concerned about the similarity between TRL and TR. The authors show some empirical results to support their intuition, however, no theoretical guarantees are provided regarding TRL superiority.  Moreover,  experiments for the Taskonomy dataset as well as on RNN have not been demonstrated, thus AR2 did not increase his/her score.  AR3 is the most critical and finds the clarity and explanations not ready for publication.

AC agrees with the reviewers in that the proposed idea has some merits, e.g. the reduction in the number of parameters seem a good point of this idea. However, reviewer urges the authors to seek non-trivial theoretical analysis for this method. Otherwise, it indeed is just an intelligent application paper and, as such, it cannot be accepted to ICLR.